# A Novel Unsupervised Video Anomaly Detection Framework Based on Optical Flow Reconstruction and Erased Frame Prediction

**DOI:** 10.3390/s23104828

**Published:** 2023-05-17

**Authors:** Heqing Huang, Bing Zhao, Fei Gao, Penghui Chen, Jun Wang, Amir Hussain

**Affiliations:** 1School of Electronic and Information Engineering, Beihang University, Beijing 100190, China; huangheqing@buaa.edu.cn (H.H.); feigao2000@163.com (F.G.); wangj203@buaa.edu.cn (J.W.); 2Inspur Electronic Information Industry Co., Ltd., Beijing 100085, China; zhaobing0412@outlook.com; 3Beihang Hangzhou Innovation Institute Yuhang, Xixi Octagon City, Yuhang District, Hangzhou 310023, China; 4Cyber and Big Data Research Laboratory, Edinburgh Napier University, Edinburgh EH11 4BN, UK; a.hussain@napier.ac.uk

**Keywords:** video anomaly detection, optical flow, incomplete event

## Abstract

Reconstruction-based and prediction-based approaches are widely used for video anomaly detection (VAD) in smart city surveillance applications. However, neither of these approaches can effectively utilize the rich contextual information that exists in videos, which makes it difficult to accurately perceive anomalous activities. In this paper, we exploit the idea of a training model based on the “Cloze Test” strategy in natural language processing (NLP) and introduce a novel unsupervised learning framework to encode both motion and appearance information at an object level. Specifically, to store the normal modes of video activity reconstructions, we first design an optical stream memory network with skip connections. Secondly, we build a space–time cube (STC) for use as the basic processing unit of the model and erase a patch in the STC to form the frame to be reconstructed. This enables a so-called ”incomplete event (IE)” to be completed. On this basis, a conditional autoencoder is utilized to capture the high correspondence between optical flow and STC. The model predicts erased patches in IEs based on the context of the front and back frames. Finally, we employ a generating adversarial network (GAN)-based training method to improve the performance of VAD. By distinguishing the predicted erased optical flow and erased video frame, the anomaly detection results are shown to be more reliable with our proposed method which can help reconstruct the original video in IE. Comparative experiments conducted on the benchmark UCSD Ped2, CUHK Avenue, and ShanghaiTech datasets demonstrate AUROC scores reaching 97.7%, 89.7%, and 75.8%, respectively.

## 1. Introduction

With the development of sensors, video surveillance and sensor networks are widely used in various fields, such as traffic monitoring, environmental monitoring, industrial control, etc. [1,2,3]. The goal of video surveillance and sensor networks is to achieve accurate monitoring and detection of anomalous targets. When analyzing real-world datasets, a common requirement is that those instances can be detected as distinct. These instances are referred to as anomalies because they are significantly different from other normal data. The anomalies may be an operation error during collecting datasets or they may originate from unknown processes that do not conform to common sense. An example of such anomaly consists of vehicles traveling on campus roads that are usually considered abnormal. The anomaly detection task is to detect these instances that do not match the expected pattern in the dataset. To accomplish this task, artificial intelligence computer vision algorithms for automatically detecting anomalies in datasets is a viable solution [4,5,6,7].

Nowadays, video surveillance is playing a more and more crucial role in smart city monitoring applications. There is a growing interest in VAD [8,9,10], which aims to identify events that do not meet normal behavior by interpreting the video. This technology can predict normal/abnormal events in a given video. Using video anomaly detection technology, smart city surveillance systems can automatically detect abnormal events, such as traffic congestion, criminal activities, fires, etc. Therefore, there are important application values in some scenes such as municipal and traffic management [11].

Deep learning approaches have made great progress in many computer vision tasks in recent years(e.g., object detection [12,13,14,15] and semantic segmentation [16,17,18,19]). In related work, many anomaly detection methods based on deep learning have been performed in different ways for detecting anomalies. Many studies have used supervised learning approaches [20,21,22] to tackle the anomaly detection task by labeling each frame of videos to clearly define the anomaly. These studies have classified normal/abnormal samples by training a convolutional neural network. However, the inter-class imbalances and intra-class imbalances in VAD interfere with the model training process; therefore, it is challenging to obtain a representative dataset of outliers, which makes the direct application of supervised learning methods difficult.

Some studies have used unsupervised learning methods for further analysis of VAD. For learning normal activities features, the usual solution is to train a model with continuous normal samples X = [$x_{1}$, $x_{2}$, $x_{3}$....$x_{n}$]. When the model is used to analyze samples in the test set, samples that do not conform to the normal mode are called anomalous samples. At present, the mainstream video anomaly detection methods based on unsupervised learning can be divided into the following two types.

(1)Reconstruction based [23,24,25]. The reconstruction errors of normal samples are tiny, but the reconstruction errors of abnormal samples are large. So, we employ an auto-encoder to reconstruct the input and set a threshold to judge whether it is abnormal.(2)Prediction based [8,26]. The model learns from the existing video frames and predicts the future frames. If there is a large difference between the prediction frame and the ground truth, the future frames are determined to be abnormal.

Although reconstruction-based/prediction-based methods have achieved good performance (more than 90% in the UCSD Ped2 dataset [27]), we consider them to have the following drawbacks.

(1)The existing studies usually process the whole image of video frames. When the objects to be detected are relatively small, they have a minor impact on the detection of abnormal events. Ideally, the precise localization of anomalous activities requires a complete representation of the subject at the object level.(2)Although the abnormal events correspond to larger reconstruction errors, due to the problem of “over-generalizing” in deep neural network, abnormal events may also be reconstructed.(3)Prediction-based methods can predict the future video frames from the given first few frames in the video sequence; then, the prediction error of a single frame can be regarded as a measurement for anomalies. However, these methods cannot make full use of the temporal context [28,29,30] and high-level semantic information of video anomaly activities.(4)Motion is the key feature for understanding videos. Current studies generally deal with images without considering the motion information in the frames. This motion information can be provided by optical flow. Optical flow can use the change of pixels in the image sequence and the correlation with adjacent frames to find the corresponding relationship between the previous frame and the current frame.

In this paper, we approach these problems from a new perspective. Instead of training with the entire video frames, we apply an object detection algorithm to extract the objects and construct STCs [31,32,33]. The STCs contain not only the objects of interest in the video frames, but also the bounding boxes of the same foreground in consecutive frames so as to eliminate the possible influence of image background. In Bert [34] and MAE [35], these models recover the occluded segments/images in a logical way according to the gradually learned context. This idea is somewhat similar to the “Cloze Test”, and we use a similar idea to construct “incomplete events” in VAD.

Specially, we erase a patch in STCs and let the model predict the erased frame according to the contextual information. Then, we extract the STCs of optical flow data and reconstruct the optical flow. Without loss of generality, we utilize an auto-encoder with storage mechanism in which the features of different layers are fused through skip connections. Motion information is used to predict the erased video frames in IE to prevent abnormal events from being reconstructed. Finally, we train the generator and discriminator to distinguish the normal mode and the prediction of optical flow. The major contributions of this paper are as follows:(1)We utilize a multi-level memory auto-encoder with skip connections to reconstruct video optical flow. By combining motion information (optical flow) and appearance information (object detection), the model can provide high-level semantics as auxiliary information to analyze the motion of video frames.(2)In order to make full use of the temporal context information of a video, we employ the idea of the “incomplete event” to predict the erased frames in the video, rather than the usual method based on future frame prediction.(3)We exploit the GAN training method and use the conditional auto-encoder as the generator. The model applies two discriminators to classify the predicted erased frames and optical flow to further improve the performance of the VAD model.(4)Finally, we conduct many experiments on the UCSD Ped2 [27], CUHK Avenue [36], and ShanghaiTech datasets [26]. The results show the advanced performance of our proposed method.

The overall structure of this paper is as follows. In Section 2, we describe the previous research on anomaly detection. Then, we design the VAD model based on hybrid tasks in Section 3. Section 4 introduces the detailed experimental setup and results of this paper. In order to analyze the effectiveness of specific components, we discuss the internal details of the model proposed in Section 5. Section 6 summarizes this paper.

## 2. Related Work

In this section, we describe different anomaly detection methods and summarize traditional machine learning methods (Section 2.1), reconstruction-based methods (Section 2.2), and prediction-based methods with various auxiliary information sources (Section 2.3).

### 2.1. Traditional Machine Learning Methods

Traditional anomaly detection models often need to design features manually to realize anomaly detection [27,37]. Early works usually extracted low-level trajectory features [38], but these approaches are not applicable in complex scenes such as occlusion. Ref. [39] proposed a hybrid model based on high-dimensional features which used the support vector machine (SVM) for normal/abnormal sample classification through unsupervised learning. The method has high scalability and computational efficiency. In addition to simple feature extraction, the help of other machine learning methods can also improve the effect of anomaly detection. Wu et al. [37] introduced chaotic dynamics into pedestrian activity events then analyzed pedestrian abnormal behavior by extracting chaotic invariant features and aggregating particle trajectories. Piciarelli et al. [40,41] analyzed the existing problems with clustering motion trajectories, which added probability information to improve the accuracy of anomaly detection. Morris et al. [42] used a hidden Markov model and maximum likelihood estimation to judge the motion route of interested nodes, which was used to forecast future activities and detect anomaly events in real-time.

### 2.2. Method of Reconstruction Error

Recent works contain many methods of deep learning, and reconstruction-based methods have become the mainstream. The basic assumption of the methods is that the reconstruction error of normal sample is low, whereas the reconstruction error of abnormal samples is high [43,44,45]. The neural network used for the reconstruction of input data is usually either an auto-encoder or GAN [46,47]. They can encode the inputs into a more compact representation and retain basic features to ensure that the reconstructed image is close to reality. Therefore, by taking advantage of auto-encoder and GAN, [24] used an auto-encoder with a convolution LSTM to model normal video frames and motion information at the same time. This method further improved the performance of the anomaly detection model. In addition, transfer learning can provide rich a priori knowledge. Salehi et al. [48] pre-trained with the ImageNet dataset to extract intermediate knowledge, then utilized the difference between the given input data and the model activation value to detect anomalies. In another work, Li et al. [49] applied a cut–paste data enhancement strategy to simulate abnormal samples and trained convolutional neural network to identify normal/abnormal samples. In the first stage, CutPaste was used to generate images for normal samples through self-supervised learning characterization. In the second stage, CNN was exploited to extract the features and calculate the anomaly score for the output features. Ref. [50] proposed a SAR image anomaly detection method based on iterative outliers and significant recursive depth, which can effectively distinguish the front background in the image and obtain an appropriate anomaly detection threshold.

### 2.3. Method of Frame Prediction

Many studies [8,51,52] have used video frame prediction methods to deal with the VAD problem. Chong et al. [53] introduced an unsupervised method to model video frames by convolution auto-encoder, which integrated data into a video representation and learned the temporal pattern of rules. The predicted video frame will significantly differ from the original frame when an abnormal event occurs. Feng et al. [54] proposed a convolutional transformer to predict future frames and used the double discriminator GAN training method to assist in the generation of the prediction frame. The model used prediction error to assist in the identification of abnormal video frames and achieved good results on different datasets. Ref. [55] expanded the Viola Jones algorithm to detect faces in videos and perform target tracking. By extracting the object level features and speed level features of the target, the model can apply these features to the classifier to complete VAD. In other frameworks, optical flow has also been used to assist RGB images for video anomaly detection. Ref. [56] recommended predicting intermediate frames in videos. The model is mainly composed of two parts: a middle frame predictor and an appearance detector, which considers the appearance and motion characteristics of video scene. Furthermore, based on the reconstructed optical flow, [33] designed a conditional variational auto-encoder to capture the correlation between video frames and optical flow field. The approach used this correlation to enhance the quality of frame prediction. Liu et al. [26] used optical flow features to assist in constraining the prediction of future frames in the video and forced the optical flow of predicted frames to be consistent with that of actual frames. Leyva et al. [57] designed a reasoning mechanism based on optical flow features and foreground key descriptive features to detect abnormal events. Ref. [58] proposed an optical flow feature clustering method with motion information to assist in anomaly detection which efficiently captured the directional motion information in the surveillance video.

## 3. The Proposed Methods

As shown in Figure 1, the pipeline proposed in this study consists of two parts: video optical flow reconstruction based on a storage mechanism and erased frame prediction based on GAN. On the one hand, we choose an object detection algorithm (Cascade R-CNN) [59] to extract local features and construct a large number of STCs for training data. This is where we explicitly focus on the object present in the video frame scene. This object detector was chosen because:(i)it uses a recurrent head network, which can efficiently detect small objects in images;(ii)the algorithm achieves a good balance between detection speed and accuracy.

By constructing STC for detected objects, then we perform optical flow extraction in them. The auto-encoder with the memory module is used to reconstruct optical flow.

On the other hand, to learn more about video temporal context information, we construct IE to predict the erased patch in STC. Two distillation methods are proposed to make the reconstructed optical flow distribution as close as possible to the normal frame distribution to achieve accurate and comprehensive positioning. Next, we add two discriminators to classify the prediction of the erased frame and the reconstruction of optical flow. The whole framework is trained on normal data. During the testing, we extract five consecutive frames from the test data then generate the erased patch to complete IE and achieve VAD. Finally, we will describe the contents of each part of the framework and represent the standards for VAD.

### 3.1. Video Optical Flow Reconstruction with Memory Mechanism

The auto-encoder consists of two parts: the encoder and the decoder. The encoder can extract low-dimensional representations from high-dimensional data, and the decoder can use the learned representations to reconstruct the original data. For VAD models, auto-encoders usually aim to minimize the represented reconstruction error as the training objective of the network.

However, the “over-generalizing” of deep neural networks often gives auto-encoders the ability to reconstruct abnormal images. Therefore, we thought of adding a memory module to the auto-encoder to design a clearer data encoding method. The method incorporates memory module into the encoder and features the data by storage factors with different weights. The model places the representations of historic normal samples, then searches for the top *k* representations with the highest similarity to the new samples representations in memory. The historic representation multiplied by a weighted sum is used to represent the samples representation. The purpose of this is that abnormal samples can only be weighted by historic normal samples representations, which makes the reconstruction error of the anomaly samples larger and easier to distinguish.

Figure 2 shows the memory module in the auto-encoder which is composed of an encoder–decoder and memory enhancement modules. The encoder consists of convolutional blocks and downsampling layers. The decoder contains upsampling layers and convolution blocks, and it is here where we add the fusion of encoder features process. Each convolution block contains a convolution layer, a batch normalization layer, and an activation layer. We use this structure to reconstruct the optical flow motion information. Firstly, the optical flow frame is encoded to obtain the output *z* and the memory module is used to convert *z* into a query $Z^{*}$. The query obtained from the memory network is shown in Formula (1).
(1)$$Z^{*}=\sum_{i=1}^{n}w_{i}m_{i}$$
where $w_{i}$ is the attention coefficient obtained from the output of the vector encoder. Each row vector $m_{i}$ represents a memory item with a dimension *C* equal to the dimension of encoder output feature. In this study, *C* is 2000. The calculation method is shown in Formulas (2) and (3). Specifically, memory is defined as a matrix containing n row vectors.
(2)$$w_{i}=\frac{exp(d\left(z,m_{i}\right))}{\sum_{j=1}^{N}exp\left(d\left(z,m_{j}\right)\right)}$$
(3)$$d\left(z,m_{j}\right)=\frac{Zm_{i}^{T}}{\left||Z|||\right|m_{i}\left|\right|}$$

To avoid the interference of unnecessary information, we use the most representative normal pattern in the top *k* record matrix. The stored values are reassigned weights and the final hidden vector $w_{i}^{*}$ is obtained for subsequent decoder operations, as shown in Formula (4).
(4)$$w_{i}^{*}=\frac{w_{i}}{max_{k}w_{i}^{*}}$$

When the top *k* with the highest similarity in memory are found in the new samples, the reconstruction error of the new samples is calculated.

To provide more information for optical flow memory, we set up a skip connection. By fusing the features obtained from different storage modules, the features of the extracted normal training data are reused. In the encoder of each memory module, at each level of the encoder, we set up a convolution block and three down-sampling layers in which the convolution block consists of a convolution layer, a batch normalization layer, and an activation layer. The storage module uses the maximum value of a similar storage slot to represent the features that are input into it, and the resulting vector represents a more informative representation of the data. For anomalous data, it is more difficult to obtain a smaller reconstruction error.

We set L2 distance as the reconstruction loss: suppose *x* is the input data and $x^{*}$ is the reconstruction result, as shown in Formula (5).
(5)$$L_{flow_{r}econ}=||x-x^{*}{\left|\right|}_{2}^{2}$$

### 3.2. Erased Frame Prediction Based on Hard Distillation

Previous studies usually complete anomaly detection based on error reconstruction and future frame prediction. These methods often cannot fully use the temporal context information of video and the high-level semantic features between adjacent frames.

In this study, we extract the training data region of interest from the video frames and construct STCs as the primary input of the model rather than the video frame of the standard method. The extracted contents are all the relevant subjects with active behavior in the frame, which make the model’s focus more individualized. Specifically, to extract the appearance information of the training data, we use the Cascade R-CNN [59] model pretrained on the COCO dataset to extract the region of interest in the video frame. Taking the extracted STCs as a set of sequences *X* = [$x_{1}$, $x_{2}$....., $x_{n}$], we use this method to mask $x_{n+1/2}$ and predict.

To better integrate appearance and motion information, we utilize the storage module to model the reconstructed of optical flow directly mapped to standard video frames. On the one hand, the reconstructed optical flow is unified into the model. On the other hand, the consistency of optical flow and video frames are encoded. The prediction accuracy is improved by trying this new mapping relationship.

Firstly, we model continuous video frames assuming that the distribution in normal mode is $p\left(\left\{x|x_{-}\left\{1:t\right\}\right\}\right)$. It is worth noting that in $x_{-}\left\{1:t\right\}$, we erase the frames in the video sequence. At the same time, given the motion information $y_{-}\left\{1:t\right\}$ of the optical flow as an auxiliary condition, it is assumed that its contents have the same hidden variable *z*.

The conditional auto-encoder is used as the generation model directly to produce $p\left(\left\{x|x_{-}\left\{1:t\right\},y_{-}\left\{1:t\right\}\right\}\right)$. Based on the implicit variable Z, we use an encoder to encode the optical flow and obtain the a priori distribution of $p\left(\left\{z|y_{-}\left\{1:t\right\}\right\}\right)$. The model uses another encoder F to process the information combining ordinary video frames and optical flow frames. The input of the encoder is the series of optical flow and normal video frames. After obtaining the posterior distribution of $q\left(z|x_{-}\left\{1:t\right\},y_{-}\left\{1:t\right\}\right)$, we sample from the posterior distribution and send it to the decoder to help predict the erased frames in IE.

We define the information distribution including the video frames and optical flow frames as the teacher network and the optical flow information distribution as the student network. In order to make the two distributions as similar as possible, we propose two distillation methods: soft distillation and hard distillation. Soft distillation can minimize the KL difference between teacher network and student network, which limits the various parts of the network as much as possible. Hard distillation distribution can deal with the global information of two distributions and take the output of teacher network as a label for student network learning. The calculation method of soft distillation is shown in Formula (6): (6)$$L_{SoftDis}=\left(1-\lambda \right)L_{CE}\left(P,y\right)+\lambda \tau^{2}KL\left(P,Q\right)$$
where *P* is $p\left(\left\{z|y_{-}\left\{1:t\right\}\right\}\right)$, *Q* is $q\left(z|x_{-}\left\{1:t\right\},y_{-}\left\{1:t\right\}\right)$. In the hard distillation distribution, we try to take the normal video frame distribution as the real label. Equation (Equation 7) is the loss function related with hard distillation.
(7)$$L_{HardDis}=L_{CE}\left(P,y\right)+L_{CE}\left(P,y_{t}\right)$$
where $L_{CE}$ is CE loss, y is the tag value, and $y_{t}$ is shown in Formula (8). These two distillation methods can allow the two distributions to learn better paradigms by learning from each other to reduce their differences.
(8)$$y_{t}=argmax_{c}Z_{t}\left(c\right)$$

In order to make the images have a better generation effect, we add a discriminator to judge the reconstruction effect of the erased frame. We use the erased frame prediction generating model as G. Specifically, G contains an encoder and a decoder. In the encoder, each downsampling block contains convolutional layers, batch normalization layers, and activation layers. The decoder consists of convolutional layers, batch normalization layers and dropout layers. For D, we employ the discriminator in PixelGan [60]; it first splices the input image and the features generated by the encoder and obtains the output through the combination of three downsampling and convolutional layers, batch normalization layers, and padding layers. The discriminator’s loss function is shown in Formula (9).
(9)$$L_{adv}^{D}=\sum_{i,j}\frac{1}{2}L_{MSE}\left(D{\left(I\right)}_{i,j},1\right)+\frac{1}{2}L_{MSE}\left(D{\left(I\right)}_{i,j},0\right)$$

During training, G and D are trained separately. The goal of training D is to judge the authenticity of the reconstructed frame. This method uses a loss function, as shown in Formulas (10) and (11). Among them, Mean Square Error (MSE) is a common method to measure the quality of the predicted image by calculating the Euclidean distance between the predicted values of all pixels in RGB color space and their real values. The goal of the training G is to generate erased frames where the weights of the discriminator are fixed. We use the MSE loss function to optimize.
(10)$$L_{adv}^{G}=\sum_{i,j}\frac{1}{2}L_{MSE}\left(D{\left(I\right)}_{i,j},1\right)$$
(11)$$L_{MSE}\left(Y^{*},Y\right)={\left(Y^{*}-Y\right)}^{2}$$

### 3.3. Criteria for Abnormal Event Detection

The test standards of VAD have always been a problem discussed by researchers. In this study, our anomaly score consists of two parts:(1)reconstruction error based on optical flow;(2)error based on frame prediction.

The abnormal fraction of optical flow is calculated as shown in Formula (12). Among them, $y_{1:t}^{*}$ and $y_{1:t}$ are the optical flow characteristics before and after reconstruction.
(12)$$S_{r}=\left|\right|y_{1:t}^{*}-y_{1:t}\left|\right|$$

In (2), we employ the predicted generated frame $I_{i}^{*}$ and the real label $I_{i}$ to calculate the error. We do not choose the L1/L2 loss function. When the input value of the function is far from the real value, the corresponding loss value is on both sides, which may cause the gradient to explode. As the literature shows, the peak signal-to-noise ratio (PSNR) is a better image quality evaluation method [61]. If the predicted video frame is normal, the method has a high PSNR value. Compared with L1/L2 loss functions, PSNR has more stable gradients and stronger structural constraints in image generation. We calculate the value of PSNR to judge whether the generated image is abnormal. The calculation formula is shown in Equation (Equation 13).
(13)$$PSNR\left(I,I^{*}\right)=10log_{10}\frac{{\left[max_{I}\right]}^{2}}{\frac{1}{N}\sum_{i=0}^{N}{\left(I_{i}-I_{i}^{*}\right)}^{2}}$$

## 4. Experiments

### 4.1. Introduction to the Datasets

UCSD ped [27]: In 2010, Mahadevan et al. established the UCSD pedestrian anomaly detection dataset. The dataset was collected by installing cameras at the height of the school to capture the scene of pedestrian streets. The abnormal behaviors that were collected by the dataset include the circulation of vehicles on the sidewalk and the occurrence of abnormal pedestrian movement, for example, cyclists, skaters, pets, etc. The UCSD dataset contains two subsets: Ped1 and Ped2. Ped1 shows how people move towards and away from the camera. There are 34 video clips in the training set and 36 video clips in the test set, with 234 × 159 pixels per frame. The Ped2 dataset is a scene where pedestrians move parallel to the camera plane. The training set and the testing set contain 16 and 14 video clips, respectively, with a resolution of 360 × 240 pixels. According to the dataset statistics, about 3400 frames include abnormal events, and the other 5500 frames are normal events.

CUHK Avenue [36]: The dataset is collection of a total of 47 anomalies in a school campus. The main anomalies include the abnormal behavior of pedestrians, the wrong direction of movement, the emergence of abnormal objects, and so on. The entire dataset contains 16 training videos and 21 testing videos, with 15,328 frames in the training set and 15,324 frames in the testing set at 640 × 360 pixels.

ShanghaiTech [26]: the production of this dataset solves the problems of the existing datasets, which are relatively focused on single aspects. The ShanghaiTech dataset collected 13 different abnormal scenes and more abnormal behavior types than the previous datasets. These scenes are shot from different angles under different lighting conditions. The main anomalies are retrograde, noise, chasing, etc. The training set contains 330 videos, and the testing set contains 107 videos, each frame with the pixels of 856 × 480.

### 4.2. Implementation Details and Evaluation Criteria

Hardware equipment. All experiments in this study were conducted on an NVIDIA RTX 2080ti GPU (Santa Clara, CA, USA) and operating system on Ubuntu 18.04. The PyTorch version used was 1.5, and the deep learning acceleration library cudnn 10.0 was used.

Implementation details. We utilized the Cascade R-CNN based on mmdetection to extract the foreground images of all datasets, forming the foreground region sequence of t frame. These regions of interest were used to build STCs. The size of each region was normalized to 32 × 32 pixels. In this study, t = 5. We used FlowNet2-SD to extract the features of optical flow and construct the STCs of optical flow according to the same method. The final distillation strategy used by the model is hard distillation. We first trained the video optical flow reconstruction model with the memory mechanism. The top value in the memory block was set to 20 to provide salient features to the auto-encoder. The batch size was initialized to 256, the learning rate was set to 0.0001, and we adopted an Adam optimizer (β1 = 0.9, β2 = 0.999). Then, we trained the generation model of the variational auto-encoder and two discrimination models. The batch size of the generation model was 128, we trained the model for 100 epochs in total. The learning rates of the generator and discriminator were initialized to 0.0001 and 0.00001, respectively, which were adjusted by the warmup strategy.

Evaluation indicators. This paper is also applicable to the popular evaluation indexes of anomaly detection models [26,50]. There are two criteria for evaluating whether a video frame is abnormal. The first criterion is the frame level standard. In this standard, if at least one pixel of a frame is marked as abnormal, the frame is regarded as abnormal. The second criterion is the pixel level standard. If at least 40 abnormal pixels are detected in a single frame, the frame is regarded as abnormal. By combining these two standards, the area under the curve (AUC) of the receiver operating characteristic curve (ROC) is calculated to measure the final performance of the model. The advantage of AUROC is that it can effectively balance the frame-level and pixel-level anomalies and eliminate the influence of threshold as much as possible. This measure directly reflects the sorting quality of the algorithm to the test samples. The higher the value, the better the result of video anomaly detection. In order to ensure the accuracy of AUROC, the model must be able to accurately find the difference between the samples tested and the training samples.

### 4.3. Experimental Result

To prove the efficiency of the method proposed in this paper, we compare our model with the SOTA model, including reconstruction-based methods such as R-VAD [45], Conv-VAD [39], MEM-VAD [62], LAD [10], GMFC-VAE [63], MemAE [64], and C-VAD [65]; future frame prediction-based methods such as VEC [66], FPVAD [26], CPNet [9], ConvVRNN [67], Attention-VAD [68], D-VAD [69], and S-VAD [70]; and methods based on multiple information sources such as ASSVAD [71], MPED-RNN [72], ST-CAE [73], AnoPCN [74], and PR-AD [75].

The results are shown in Table 1. It can be observed that our method achieves excellent performance on all three datasets when compared to the state-of-the-art methods, which demonstrates the advancements made by our method. Specifically, this method can reach a 97.7% AUROC score in the Ped2 dataset. In the above literature, the best VAD result for a reconstruction-based method is 95.1%, the best future frame prediction-based [66] method’s result is 97.3%, and the best method that integrates multiple information sources reached a result of 97.2%. Our method is higher than the results of these methods. In the CUHK Avenue dataset, our method provided an absolute gain of 0.4%, 0.1%, and 3.3% over the best results of the above three methods, respectively. The VAD result of this paper on the ShanghaiTech dataset is 75.8%, which is higher than 74.8% in the VEC [66] literature. We believe that this is the effect of the combination of optical flow assistance and temporal context information. Our pipeline combines the advantages of the above two parts and achieves a better video anomaly detection effect.

Figure 3 shows the effect of VAD on the UCSD Ped2 dataset. We visualized a two-frame video sequence. In the drawn graph, the abscissa is the video frame sequence, whereas the ordinate is the abnormal response value. The red areas represent the real label of the anomaly, and the black curves represent the response of the model proposed in this study. In Figure 3a, “retrograde” is identified as an abnormal event on the pedestrian street. When the video is played to approximately frame 83, a pedestrian is moving retrograde to the camera. This behavior is different from most behaviors in the normal frame, and the exception value increases accordingly. In Figure 3b, at about frame 25, “cyclist” appears in the scene. The model can effectively detect this as an anomaly.

We show the VAD effect on the CUHK Avenue dataset in Figure 4. The abnormal behaviors in Figure 4a,b are “the person throwing the schoolbag” and “the person pushing the bicycle”.

### 4.4. Ablation Experiment

To analyze the effects of different components of the model proposed in this study, we conducted ablation experiments on the UCSD Ped2 and CUHK Avenue datasets and evaluate it with the AUROC evaluation index. The results are shown in Table 2. Our model is divided into three components: optical flow reconstruction based on the memory module, intermediate frame prediction, and multi-discriminator optimization. The baseline we used is a simple prediction of future frames model. The ablation experiments were carried out by accumulating these three parts, respectively. In this experiment, the distillation method for optical flow distribution and overall distribution is hard distillation.

It can be seen from the table that the prediction method with only future frames is 94.1%. A 95.23% AUROC score was obtained by adding optical flow motion information with the memory module. Adding optical flow information can make the model more sensitive to abnormal activities. After using video context information, the prediction result of the erased frame is 96.07%; this shows that rich temporal information is crucial for improving model accuracy. The final result is 97.7% after adding multiple discriminators. We believe that this multi-discriminator pipeline is beneficial to balance the training of the network and improve the quality of the generated images. On the CUHK Avenue dataset, our method was 3.4% higher than the original baseline. All the methods are superior to the previous methods that can only predict future frames, which reflects the superiority of the method proposed in this study.

We show the abnormal score images of different components on the Ped2 dataset in Figure 5. In Figure 5d, although the method of using only future frame prediction can achieve results, the score value of abnormal behavior is low in about 100–160 frames. In Figure 5c, by erasing frames, we utilize temporal context information to assist prediction. Then we use GAN’s training method, which can obtain more response. Figure 5b shows the prediction of the optical flow reconstructed by the memory module. In Figure 5a, we combine all the information to test. Around frames 100–120, our model can detect obvious abnormalities, and the detection effect is smoother than others.

Similarly, we also performed ablation experiments on the CUHK Avenue dataset, and the abnormal score image of each part is shown in Figure 6. When we combined optical flow information and temporal context information, the model was more sensitive to anomaly detection. When an abnormal event had just occurred, the response of the model was also stronger than others.

In Figure 7, we visualize a heatmap of anomaly detection results on two datasets. The heatmap contains rich color changes and full information, which helps us add color representations of detected objects to the image. In the VAD task, we use the heatmap to indicate which areas are abnormally high-risk areas. In Figure 7a, we show that the anomalies are “cyclists” and “skateboarders”, and their trajectories can be clearly distinguished. In Figure 7b, the abnormal event we visualized is a “runner”. When an abnormal event occurs, its activity will be more intuitively represented.

## 5. Discussion

### 5.1. Top *k* in Memory Mechanism

We attempted to experiment with the memory mechanism of optical flow reconstruction, and the experimental results on the two datasets are shown in Table 3. In Ped2 dataset, through the sigmoid and weighted sum of the weights in all dimensions, the AUROC score was 95.6%. This method generates a lot of computational effort and affects the speed of model training. Next, considering the sparse addressing method, we set different top *k* values for experimental analysis. When K was 5, 10, 20, and 30, the final AUROC values were 95.4%, 95.9%, 97.7%, and 96.9%, respectively. When K was 20, our model’s result was the best. In the CUHK Avenue dataset, an AUROC score of 89.3 % was obtained using the weighted average method. When K = 20, the AUROC value was still the highest, at 89.7%. The sparse addressing method enables the model to learn the representation of data with less but more relevant memory values. The latent space obtained by the encoder is used to retrieve the most relevant weight of the memory module to minimize the reconstruction error.

### 5.2. Different Numbers of Temporal Context Information

In order to better complete incomplete events, we analyzed the number of contexts in the erased frames. The experiment was carried out on two datasets, and the results are shown in Table 4. Firstly, the video frames were preprocessed and the $t_{th}$ frames were extracted as video sequences. In Ped2, when t = 7 and the number of front and back contexts was three frames, the model obtained a resulting AUROC score of 96.6%. When t = 5 and the number of front and back contexts was 2, the model obtained an AUROC score of 97.7%. When t = 3, the AUROC score was only 94.2%. In the CUHK Avenue dataset, the resulting AUROC score was only 81.0% when t = 3. When t = 5, the best result was 89.7%. The analysis results show that more context information may provide redundant information. Too little context cannot provide enough temporal information to effectively complete VAD tasks.

### 5.3. Different Distillation Methods

Knowledge distillation can compress the information learned by a teacher network into the student network. We transfer this learning inductive bias method to the optical flow model and take the normal video frame model as the teacher network. At the same time, we carry out experiments on the two distillation methods proposed in this study, and the results are shown in Table 5. In the Ped2 dataset, the final result obtained by the hard distillation method was 0.2% higher than that achieved by the soft distillation method. In the CUHK Avenue dataset, the performance is more obvious, and the model’s effect was improved by 1.6%. Using the hard distillation method can make the student network more dependent on the hard labels provided by the teacher network. We think that this method can make the model learn the global information more effectively. The experiments showed that this hard distillation method can more effectively restore the original video of IE with the help of optical flow motion information.

### 5.4. Differing Numbers of Model Parameters

The number of different model parameters can directly affect the performance and efficiency of the model. On the Ped2 dataset, we compared the number of parameters of the model proposed in this paper with three other methods, and the results are shown in Table 6. The number of parameters of the proposed method reached 287 M, which is higher than in MemAE and HF2VAD. We believe that this is brought on by the computation of optical flow and the GAN training method, both of which create a large number of parameters to be computed. In our future work, we will reduce the number of model parameters as much as possible while maintaining accuracy and reducing the storage and computational costs of the model.

## 6. Conclusions

In this work, we combined the methods of reconstruction-based/frame prediction-based to obtain a high-precision VAD model. Our design idea is novel, predicting future frames by utilizing a variational auto-encoder as a generator while taking previous video frames and optical flow as inputs, and it proposes an efficient reconstruction method, i.e., an auto-encoder with memory structures. Instead of simply combining reconstruction and prediction, our architecture constrains the reconstructed optical flow and video frames by designing different distillation methods. At the same time, we make full use of the temporal information in the video to make predictions by erasing a patch in consecutive video frames. To generate high-quality predictions, we also add two discriminators to ensure that the generated predictions are consistent with the ground truths. This way, we can guarantee that events that are normal in terms of appearance and motion are identified as such, whereas events predicted to differ significantly from the ground truth are classified as anomalous. To verify the validity of this method, we performed experiments on three different datasets. The AUROC score achieved by our model on the UCSD Ped2 datasets was 97.9%, and those of the CUHK Avenue and ShanghaiTech datasets reached 89.7% and 75.8%, respectively. We performed ablation experiments with the first two datasets, achieving performance increases of 3.6% and 3.4%, respectively, when compared with the baseline. Our experiments show that this method can effectively manage VAD tasks. The results of this paper can play an important role in practical applications such as smart city surveillance and security fields and provide new ideas and methods to achieve more accurate and efficient target anomaly detection.

## Figures and Tables

**Figure 1 sensors-23-04828-f001:**
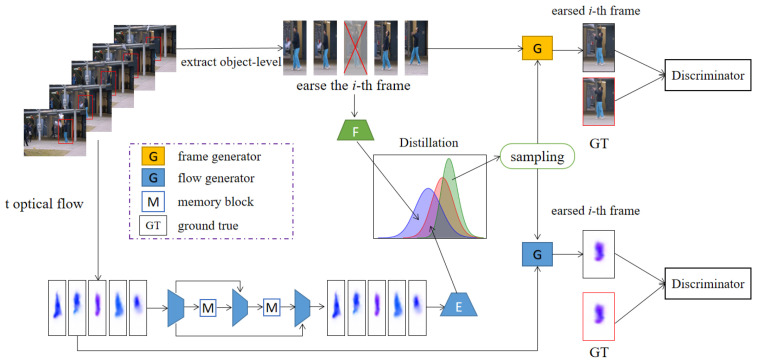
The anomaly detection framework proposed in this study includes the following contents. Firstly, we mask the patch in the video frame sequence. Then, the video sequence is fed to an optical flow network to extract motion information. The optical flow is reconstructed through an auto-encoder with the storage module. We sample from the distribution and input it into the variational auto-encoder to generate the prediction of the erased frame. Finally, we use a discriminator to classify it.

**Figure 2 sensors-23-04828-f002:**
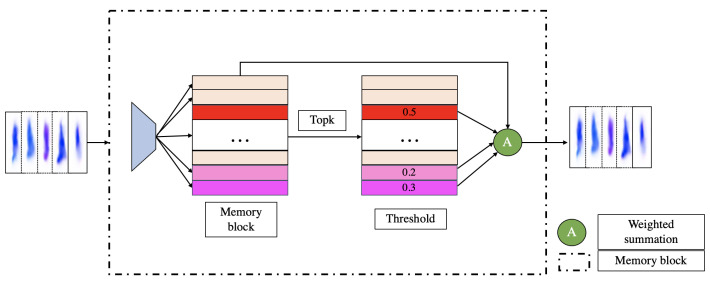
Structure of auto-encoder with memory modules. This memory block appears multiple times in the structure, and we fuse the blocks using skip connections to provide richer information to the model.

**Figure 3 sensors-23-04828-f003:**
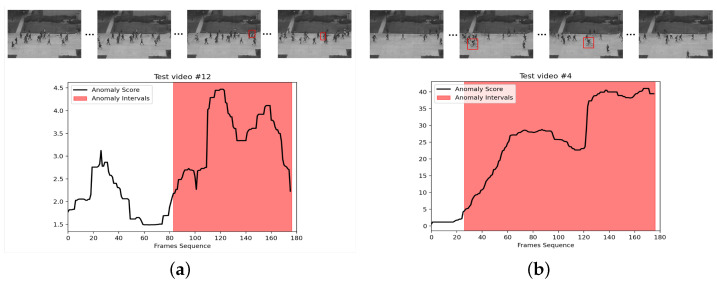
We show two abnormal events on the UCSD Ped2 dataset. The red border in the diagram represents abnormal behavior. The red area is a part of the real label of the anomaly, and the black curve represents the score of the anomaly. The higher the value, the more likely the exception will occur. (**a**) On a normal street from left to right, a pedestrian turned retrograde; (**b**) At frame 25 of the video, a man on a bicycle appeared in the street where pedestrians were walking.

**Figure 4 sensors-23-04828-f004:**
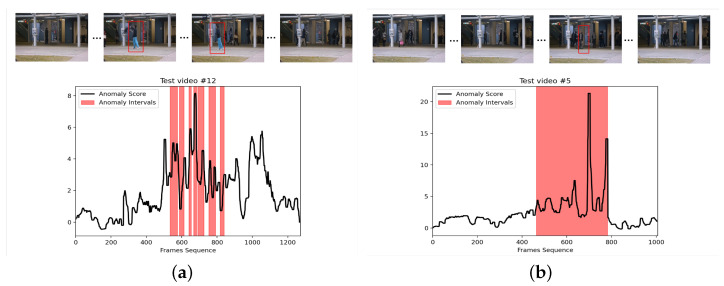
Two anomaly examples from the CUHK Avenue dataset. (**a**) A man in blue trousers scattered pieces of paper on the road. (**b**) In frames 500 to 780 of the video, a man walks out with a bicycle.

**Figure 5 sensors-23-04828-f005:**
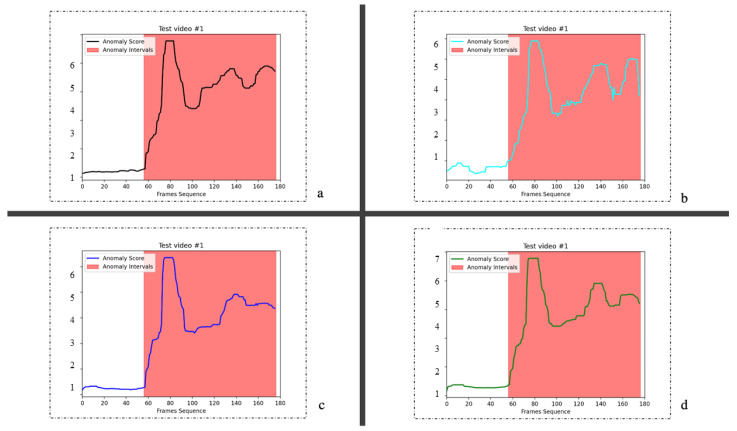
The different structures in the ablation experiment are analyzed on the Ped2 dataset, in which (**a**) is our overall model, (**b**) is the optical flow reconstruction model based on the memory module, (**c**) is the prediction of the erased frame with two discriminators, and (**d**) is the prediction of the original future frame.

**Figure 6 sensors-23-04828-f006:**
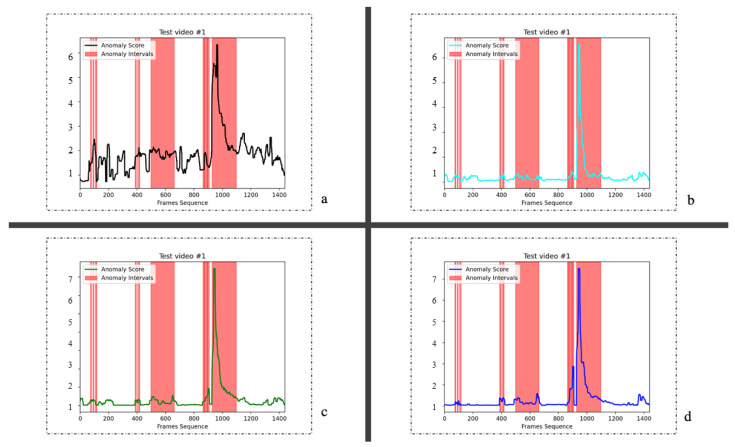
Test effects of different components of the model in the CUHK Avenue dataset, all legends are the same as shown in Figure 5.

**Figure 7 sensors-23-04828-f007:**
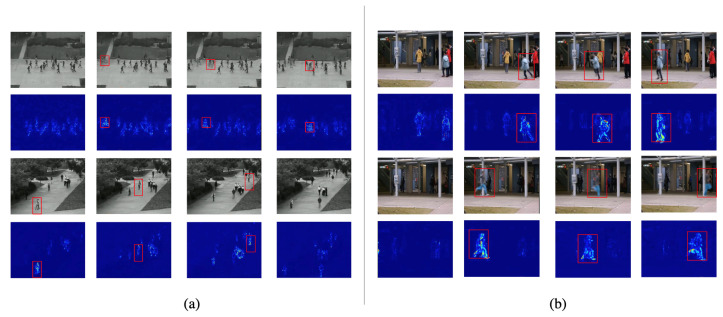
Visual example of frame prediction comparison In different datasets, we show normal and abnormal data. The first and third rows are the raw video frames in the dataset, and the second and fourth rows are the heatmaps of the frames. From left to right, we show (**a**) the Ped2 dataset, where the anomalies are “bicyclist” and “skateboarder” and (**b**) the CUHK Avenue dataset, where the anomalies are all “running”. Brighter colors in the heatmap indicate larger prediction errors.

**Table 1 sensors-23-04828-t001:** Video abnormal detection results (in %) in terms of AUROC score on the UCSD Ped2, CUHK Avenue, and ShanghaiTech datasets. Our method is compared with various start-of-the-art approaches, which are sorted by type of method.

Category	Method	Ped2	CUHK Avenue	ShanghaiTech
	R-VAD [45]	92.2	81.7	68.0
	Conv-VAD [39]	90.0	70.2	60.9
	MEM-VAD [62]	90.2	82.8	69.8
Restructure	LAD [10]	95.1	89.3	/
	GMFC-VAE [63]	92.2	83.4	/
	MemAE [64]	94.1	83.3	69.8
	C-VAD [65]	87.5	84.4	
	VEC [66]	97.3	89.6	74.8
	FPVAD [26]	95.4	85.1	72.8
	CPNet [9]	96.1	85.1	/
Prediction	ConvVRNN [67]	96.1	85.8	/
	Attention-VAD [68]	96.0	86.0	/
	D-VAD [69]	95.6	84.9	73.7
	S-VAD [70]	/	89.6	74.7
	ASSVAD [71]	96.7	86.4	71.6
	MPED-RNN [72]	/	/	73.4
Hybrid Frame	ST-CAE [73]	91.2	80.9	/
	AnoPCN [74]	96.8	86.2	73.6
	PR-AD [75]	96.3	85.1	73.0
	Ours	**97.7**	**89.7**	**75.8**

**Table 2 sensors-23-04828-t002:** We analyzed different components on the UCSD Ped2 and CUHK Avenue dataset. By superimposing the baseline, the contribution of each component to the model is evaluated.

Dataset	Baseline	Optical Flow	Erased Frame Prediction	Multi-Discriminator	AUROC (%)
	✓				94.1
	✓	✓			95.2
Ped2	✓	✓	✓		96.1
	✓	✓	✓	✓	97.7
	✓				86.3
	✓	✓			87.2
CUHK Avenue	✓	✓	✓		87.6
	✓	✓	✓	✓	89.7

**Table 3 sensors-23-04828-t003:** In the module of memory reconstruction optical flow, we analyze the influence of different top *k* on the model results.

Dataset	Top *k*	AUROC (%)
	K = 5	95.4
	K = 10	95.9
Ped2	K = 20	97.7
	K = 30	96.9
	Weight mean	95.6
	K = 5	88.1
	K = 10	88.6
CUHK Avenue	K = 20	89.7
	K = 30	89.1
	Weight mean	89.3

**Table 4 sensors-23-04828-t004:** We compared the effects of different numbers of before and after frames on intermediate frame prediction.

Dataset	t = ?	AUROC (%)
	t = 3	94.2
Ped2	t = 5	97.7
	t = 7	96.6
	t = 3	81.0
CUHK Avenue	t = 5	89.7
	t = 7	88.9

**Table 5 sensors-23-04828-t005:** We carried out experiments on different distillation methods.

Dataset	Soft Distillation	Hard Distillation	AROUC (%)
Ped2	✓		97.5
		✓	97.7
CUHK Avenue	✓		88.1
		✓	89.7

**Table 6 sensors-23-04828-t006:** Comparison of different method parameter quantities.

Model	Parameters (M)
MemAE [64]	22.8
HF2VAD [33]	252
FPVAD [26]	339
Ours	287

## Data Availability

Not applicable.

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
