# Peer review of "A Novel Unsupervised Video Anomaly Detection Framework Based on Optical Flow Reconstruction and Erased Frame Prediction"

_sensors, 2023, doi:10.3390/s23104828_

Round 1
Reviewer 1 Report
The paper proposes a novel unsupervised video anomaly detection (VAD) framework based on optical flow reconstruction and erased frame prediction. The authors argue that traditional reconstruction and prediction-based approaches are insufficient to utilize the rich contextual information in videos, which makes it difficult to perceive anomaly activities accurately. They propose a Close Test strategy inspired by natural language processing (NLP) to encode both motion and appearance information at an object-level. The proposed framework includes an optical stream memory network, a space-temporal cube (STC), and a conditional autoencoder. The authors employ a Generative Adversarial Network (GAN) based training method to improve the performance of VAD.
Strengths:
The proposed framework achieves state-of-the-art performance on three benchmark datasets, UCSD ped2, Avenue, and ShanghaiTech. The AUROC scores reach 97.7%, 89.7%, and 75.8%, respectively. The proposed framework is novel and innovative, as it combines different approaches, such as optical flow reconstruction, erased frame prediction, and GAN-based training. The authors provide detailed experimental results and analysis, demonstrating the effectiveness of their proposed framework.
Weaknesses:
The paper lacks a clear motivation for the proposed framework, as the authors do not discuss the limitations of existing methods and the specific challenges they aim to address. The paper could benefit from more thorough literature review and discussion, particularly in the area of scene graphs and zero-shot learning. Additionally, the authors could provide more details on the implementation of their framework, such as the hyperparameter settings and the training procedure.
Citations:
The authors could consider citing the following papers:
-
"A Comprehensive Survey of Scene Graphs: Generation and Application" - This paper provides a comprehensive survey of scene graph generation, which is related to the proposed framework as it involves encoding contextual information in images.
-
"TN-ZSTAD: Transferable Network for Zero-Shot Temporal Activity Detection" - This paper proposes a zero-shot learning method for temporal activity detection, which could be relevant to the proposed framework.
-
"Video Pivoting Unsupervised Multi-Modal Machine Translation" - This paper proposes an unsupervised video translation framework, which is related to the proposed framework as both aim to utilize the rich contextual information in videos.
-
"ZeroNAS: Differentiable Generative Adversarial Networks Search for Zero-Shot Learning" - This paper proposes a differentiable GAN-based approach for zero-shot learning, which is relevant to the proposed framework as it employs a similar training method.
Justifications:
The above papers could provide additional context and insights for the proposed framework. The first paper provides a comprehensive survey of scene graph generation, which could be relevant to the encoding of contextual information in videos. The second paper proposes a zero-shot learning method for temporal activity detection, which could be relevant to the proposed framework. The third paper proposes an unsupervised video translation framework, which is related to the proposed framework as both aim to utilize the rich contextual information in videos. The fourth paper proposes a differentiable GAN-based approach for zero-shot learning, which is relevant to the proposed framework as it employs a similar training method.
Author Response
Reply to Comments of Editors and Reviewers
A Novel Unsupervised Video Anomaly Detection Framework Based on Optical Flow Reconstruction and Erased Frame Prediction
Heqing Huang, Bing Zhao, Fei Gao, Penghui Chen, Jun Wang, and Amir Hussain
Dear Editors and Reviewers:
We would like to thank the editors and the reviewers for their invaluable comments on our manuscript entitled “A Novel Unsupervised Video Anomaly Detection Framework Based on Optical Flow Reconstruction and Erased Frame Prediction”. We have tried our best to address all comments carefully and made corresponding corrections which have improved the quality of the manuscript.
Review 1:
Commend 1: The authors could consider citing the following papers:
"A Comprehensive Survey of Scene Graphs: Generation and Application" - This paper provides a comprehensive survey of scene graph generation, which is related to the proposed framework as it involves encoding contextual information in images.
"TN-ZSTAD: Transferable Network for Zero-Shot Temporal Activity Detection" - This paper proposes a zero-shot learning method for temporal activity detection, which could be relevant to the proposed framework.
"Video Pivoting Unsupervised Multi-Modal Machine Translation" - This paper proposes an unsupervised video translation framework, which is related to the proposed framework as both aim to utilize the rich contextual information in videos.
"ZeroNAS: Differentiable Generative Adversarial Networks Search for Zero-Shot Learning" - This paper proposes a differentiable GAN-based approach for zero-shot learning, which is relevant to the proposed framework as it employs a similar training method.
Many thanks for the insightful suggestions. We added these documents, which can be viewed in references 28, 29, 30, and 47.
Reviewer 2 Report
The article proposes a framework for unsupervised video anomaly detection. I have the following comments:
1. Line 4- I believe that u mean the "Cloze test" not close
2. I would like a comparison to be added as to the size of the compared model versus to yours in terms of number of parameters or flops.
The authors presented a novel method to detect anomalies in videos and assessed this method using an NLP assessment concept. The evaluation indicator used are the most common considering the anomaly detection and produced results almost similar to the highest model on each dataset which suggests the generalization ability of the proposed model. Finally an ablation study was also presented. I think the paper is very well written and the flow is very good
You have some English mistakes, kindly revise
Author Response
Reply to Comments of Editors and Reviewers
A Novel Unsupervised Video Anomaly Detection Framework Based on Optical Flow Reconstruction and Erased Frame Prediction
Heqing Huang, Bing Zhao, Fei Gao, Penghui Chen, Jun Wang, and Amir Hussain
Dear Editors and Reviewers:
We would like to thank the editors and the reviewers for their invaluable comments on our manuscript entitled “A Novel Unsupervised Video Anomaly Detection Framework Based on Optical Flow Reconstruction and Erased Frame Prediction”. We have tried our best to address all comments carefully and made corresponding corrections which have improved the quality of the manuscript.
Review 2:
Commend 1: Line 4- I believe that u mean the "Cloze test" not close.
Many thanks for the insightful suggestions. We have modified the specific formulation in line 4 by changing "Close Test" to "Cloze Test".
Commend 2: I would like a comparison to be added as to the size of the compared model versus to yours in terms of number of parameters or flops.
Many thanks for the insightful suggestions. We have added a discussion section by comparing the number of parameters of the proposed model in this paper with the other three models:
The number of different model parameters can directly affect the performance and efficiency of the model. On the ped2 dataset, we compared the number of parameters of the model proposed in this paper with three other methods, as shown in Table 6. The number of parameters of the proposed method reaches 287M, which is higher than MemAE and HF2VAD. We believe this is brought by the computation of optical flow and the training method of GAN, both of which bring a large number of parameters to be computed. In our future work, we will reduce the number of model parameters as much as possible while maintaining the accuracy and reducing the storage and computational cost of the model.
Model |
Parameters(M) |
MemAE |
22.8 |
Hf2VAD |
252 |
FPVAD |
339 |
Ours |
287 |
Table 6. We compare different method parameter quantities.